# In Vitro and In Vivo Antibiofilm Activity of Red Onion Scales: An Agro-Food Waste

**DOI:** 10.3390/molecules28010355

**Published:** 2023-01-01

**Authors:** Nermeen B. Ali, Riham A. El-Shiekh, Rehab M. Ashour, Sabah H. El-Gayed, Essam Abdel-Sattar, Mariam Hassan

**Affiliations:** 1Department of Pharmacognosy, Faculty of Pharmacy, Cairo University, Cairo 11562, Egypt; 2Department of Pharmacognosy, Faculty of Pharmacy, 6th October University, Cairo 12585, Egypt; 3Department of Microbiology and Immunology, Faculty of Pharmacy, Cairo University, Cairo 11562, Egypt; 4Department of Microbiology and Immunology, Faculty of Pharmacy, Galala University, New Galala City, Suez 43511, Egypt

**Keywords:** agro-waste, *Allium cepa*, antibacterial, MRSA, vaginal colonization, standardization

## Abstract

Red onion wastes (ROW) are valuable sources of bioactive metabolites with promising antimicrobial effects. Methicillin-resistant *Staphylococcus aureus* (MRSA) infections are a growing risk in hospitals and communities. This study aims to investigate the in vitro and in vivo antibiofilm activities of the acidified ethanolic extract of red onion scales (RO-T) and its fractions against an MRSA vaginal colonization model. The RO-T extract, as well as its anthocyanin-rich fraction (RO-P) and flavonoid-rich fraction (RO-S), recorded a promising antibacterial activity against highly virulent strains of bacteria (MRSA, *Acinetobacter baumannii*, *Escherichia coli* and *Pseudomonas aeruginosa*). RO-S showed the highest antibacterial activity (MBC of 0.33 ± 0.11 mg/mL) against MRSA USA300 and significantly eradicated its biofilm formation with an IC_50_ of 0.003. Using a rat model, in vivo assessment on all samples, which were formulated as a hydrogel, revealed a significant reduction of MRSA bacterial load recovered from an infected vagina compared to that of the negative control group (NCG). RO-T extract and vancomycin groups recorded the highest antibacterial activity with a bacterial load 2.998 and 3.358 logs lower than the NCG, respectively. The histopathological investigation confirmed our findings. RO-T and RO-S were standardized for their quercetin content. Finally, ROW offers a new potent antibiofilm agent mostly due to its high quercetin content.

## 1. Introduction

Onion (*Allium cepa* L.) is one of the most cultivated vegetables in the world. It is rich in abundant phytoconstituents which aid in the cure and prevention of a high number of illnesses [1]. Onions of red, yellow, and white colors are known for their bioactive phenolic compounds, such as organosulphur compounds, flavonoids, phenolics, and anthocyanins [2]. The non-edible outer layers of the onion are a widely available waste product from the onion’s processing and utilization either at the household, kitchen, or industrial level [3]. As consumption expands, the generation of onion wastes increased. Thus, efficient environmental disposal and usage of those waste products is required [4].

Red onion outer dry scales are rich sources of natural phytoconstituents such as anthocyanins and flavonoids. Several studies confirmed their higher flavonoid content compared to the edible part of the plant; these flavonoids include quercetin, kaempferol, luteolin, and other quercetin derivatives [5]. Quercetin and its derivatives have potential antioxidant, antiallergic, anti-inflammatory, and antimicrobial activities and a broad-spectrum antibiofilm effect against various bacterial pathogens such as *Enterococcus faecalis*, *Staphylococcus aureus,* and *Pseudomonas aeruginosa* [6]. Throughout the years, human beings have suffered from many severe and complex diseases caused by various microorganisms. Nevertheless, several microorganisms have recently acquired resistance against antibiotics and different synthetic and natural antibacterial agents due to the abuse of these drugs.

According to the WHO (World Health Organization), antimicrobial resistant bacteria are considered to be clinically most impactful to human health and have been assigned for observation [7]. *Staphylococcus aureus* is a common opportunistic pathogenic bacterium that causes nosocomial and community-acquired infections in humans. Methicillin is one of the antibiotics that act against beta-lactamase-producing *Staphylococci*. The misuse of methicillin led to the development of the methicillin-resistant *Staphylococcus aureus* (MRSA) strain that is associated with an increase in serious human illnesses. MRSA causes a wide variety of infections ranging from skin and soft tissue infections to serious illnesses such as septicemia, infective endocarditis, and pneumonia [8,9]. Thus, the need to discover new, natural, potent antimicrobial agents has been increased [10]. If these bacteria are carried in the female genital tract, they could be transmitted to the neonate, resulting in local or systemic neonatal infections that will be difficult to treat with commonly available antimicrobials. In order to develop new effective treatment strategies, there is demand for updated data about the prevalence of colonization with significant antimicrobial-resistant pathogens. The present study aims to evaluate the antibiofilm activity of the total extract of red onion dry outer scales and its fractions against MRSA vaginal colonization and, finally, to support the market with formulated standardized products as valuable tools for multidrug resistant infections.

## 2. Results

### 2.1. Screening of the Antibacterial Activity

The antibacterial activity of the tested extract/fractions was determined using minimum bactericidal concentration (MBC). All the tested extract/fractions showed bactericidal activity against all the tested pathogens, except RO-P fraction did not show any bactericidal activity against *Pseudomonas aeruginosa PAO1* (Table 1). The highest antibacterial activity was recorded against MRSA USA300 with a minimum bactericidal activity of 0.65 ±0.23, 4.17 ±1.8, and 0.33 ±0.11 mg/mL for RO-T, RO-P, and RO-S samples, respectively.

### 2.2. Effect of the Tested Extracts on the Biofilm Activity

The tested extract/fractions significantly inhibited MRSA USA300 and *A. baumannii* AB5057 biofilm formation at all the tested concentrations. RO-T extract also showed significantly greater antibiofilm activity against *A. baumannii* AB5057 compared to MRSA USA300 (Two-way ANOVA, Sidak’s multiple comparisons test, *p* < 0.0001) at all the tested concentration levels (Figure 1). Concerning the RO-P fraction, it showed significantly higher antibiofilm activity against *A. baumannii* AB5057 than against MRSA USA300 (Two-way ANOVA, Sidak’s multiple comparisons test, *p* < 0.0001) at a concentration of 2 mg/mL (Figure 1). All the tested extract/fractions showed significantly higher biofilm eradication activity against *A. baumannii* AB5057 than against MRSA USA300 (Two-way ANOVA, Sidak’s multiple comparisons test, *p* < 0.0001) at all the tested concentrations (Figure 1). The recorded mean IC_50_ (half maximal biofilm inhibitory concentration) and EC_50_ (half maximal biofilm eradication concentration) against the *A. baumannii* AB5057 biofilm for the RO-T, RO-P, and RO-S samples were higher than that against the MRSA USA300 biofilm (Table 2). The RO-P extract recorded the highest IC_50_ against the biofilms of both MRSA USA300 and *A. baumannii* AB5075 (Table 2).

### 2.3. In Vivo MRSA Vaginal Colonization Model

The promising in vitro antibacterial and anti-virulence activity of the tested extract/fractions was validated by investigating the in vivo activity through a rat model of MRSA vaginal infection (Figure 2). All the tested extract/fractions significantly reduced the MRSA bacterial load recovered from the infected vagina compared to that of the negative control (no treatment) group (Kruskal–Wallis Test, Dunn’s post-hoc test, *p* < 0.05). The RO-T extract and vancomycin groups recorded the highest antibacterial activity in vivo with a bacterial load 2.998 and 3.358 logs lower than that of the negative control group, respectively. There was no significant difference between the bacterial loads recovered from all the tested-extract-/fractions-treated groups and the vancomycin-treated group (positive control) (Kruskal–Wallis Test, Dunn’s post-hoc test, *p* < 0.05) (Figure 2). In addition, there was no significant difference between the bacterial loads recovered from the high concentration (50 mg/mL) or the low concentration (12.5 mg/mL) of the tested-extract-/fractions-treated groups (Kruskal–Wallis Test, Dunn’s post-hoc test, *p* < 0.05) (Figure 2).

### 2.4. Histopathological Analysis

Microscopic examination of group 1 (Healthy) revealed normal structure of the vaginal wall without any detectable alterations. Group 2 (Infection+ vehicle) showed mild vacuolation in the epithelial lining of the vagina, with inflammatory cell infiltration in the lamina propria. Group 3 (Bacterial infected) exhibited mild edema with inflammatory cell infiltration in the lamina propria. Group 4 (Vancomycin) showed apparently normal vaginal mucosa except for a few focal aggregations of inflammatory cells seen in the lamina propria. Group 5 (RO-T, 12.5) exhibited a moderate inflammatory reaction in the lamina propria of the vaginal wall. Group 6 (RO-T, 50) showed apparently normal vaginal wall with the presence of few congested blood vessels. Group 7 (RO-P, 12.5) exhibited congested blood vessels with mild perivascular edema and subsided inflammation. Group 8 (RO-P, 50) showed similar histological picture regarding mild edema and congestion. Group 9 (RO-S, 12.5) showed mild inflammatory edema. Group 10 (RO-S, 50) exhibited apparently normal mucosa with congested blood vessels (Appendix A).

### 2.5. HPLC Standardization

The HPLC chromatograms of the (1% HCL) acidified ethanolic extract of the red onion scales RO-T and its flavonoid-rich fraction RO-S (the most active) demonstrated a major peak for quercetin detected at 13.2 and 13.1 min, respectively. The peak of quercetin was identified and confirmed by comparison with HPLC chromatogram of the reference quercetin standard (Figure 3). According to the previously established HPLC calibration curve, it can be concluded that each 1 mg of acidified ethanolic extract contains 0.0478 ± 0.0009 mg quercetin, and each 1 mg of flavonoid-rich fraction contains 0.2188 ± 0.0036 mg quercetin.

## 3. Discussion

The development of multidrug resistance between clinical pathogens has been reported to be coupled with biofilm formation [11]. The predominance of *Staphylococcus* resistant bacteria in hospital and community infections, especially in immunocompromised patients, has motivated many researchers to investigate these bacteria. Genus *Staphylococcus* bacteria are known to be able to develop resistance against many antibiotics, which could delay the patient’s cure and cause high morbidity and mortality rates. *Staphylococcus aureus* is the major cause of bacterial infections in the genus *Staphylococcus*. The vancomycin antibiotic is the only one that can be used against infections of methicillin-resistant *Staphylococcus* strains. Recently, it showed inefficiency in some cases, which caused fear among healthcare professionals around the globe [12]. Hence, novel antimicrobial agents that cannot be countered by such pathogens are essentially needed [11]. Utilization of naturally occurring constituents that are derived from plant sources for the cure of different human disorders such as resistant bacterial infections is a promising safe alternative to using harmful synthetic drugs.

Onion is the most important horticultural crop cultivated globally. Onion processing produces numerous wastes, mainly the outer scales. Many researchers have explored onion outer scales as a rich source of antioxidants, dietary fibers, fructo-oligosaccharides, and polyphenols [13]. It has been reported that onion outer scales contain a significantly higher flavonoid content as compared to the edible part [14]. There are two main flavonoid subclasses present in red onion: anthocyanins and flavonol aglycone (quercetin type) and their derivatives, which all contribute to the yellow to purple color of onion scales [15]. On the way to relating the assessed activity with active metabolites, we extensively separated the flavonoids from the anthocyanins via the ether precipitation method and investigated both fractions alongside with the total extract. This is the first study concerning the inhibitory potential of red onion outer scales against MRSA vaginal colonization.

The evaluation of antibacterial effect against different pathogens revealed that the flavonoid-rich fraction (RO-S) has the highest bactericidal activity, especially against MRSA USA300. This is in accordance with the research previously reported about quercetin (the major flavonoid in the RO-S fraction) from different medicinal plants, which showed its potent antibacterial activity against a wide range of bacterial strains. This action has been linked to its solubility and its interaction with the bacterial cell membrane. In addition, quercetin has been well documented to have broad-spectrum antibacterial activities against Gram-positive bacteria such as methicillin-resistant *Staphylococcus aureus* (MRSA) [16].

It is noteworthy that the antimicrobial agents commonly used to treat vaginal colonization are usually ineffective, causing high failure rates, and this has often been associated with biofilm formation [17]. Furthermore, the in vitro and in vivo experiments revealed that the total extract and its flavonoid-rich fraction (RO-S) are powerful antibiofilm agents in comparison to the anthocyanin-rich fraction (RO-P), which also showed significant activity. In addition, the study revealed that red-onion outer-scales ethanolic extract and its fractions have a dose-dependent antibiofilm effect. The larger doses showed higher biofilm inhibition and biofilm eradication activities in vitro. Moreover, there was a great reduction of the bacterial load recovered from the infected vaginal tissue in vivo, which was confirmed by the histopathological study. The potential activity of the total ethanolic extract may be attributed to the synergistic effect of all its bioactive metabolites, viz, anthocyanins, phenolic acids, and flavonoids, especially quercetin.

Several studies investigated the chemical profile of red onion peels, and the authors documented that the polyphenols dominated metabolites in the extract as ferulic acid; 2-(3,4-dihydroxybenzoyl)-2,4,6-trihydroxy-3(2H)-benzofuranone; protocatechuic acid; quercetin; quercetin 4′-O-α-D-glucopyranoside; quercetin dimer; quercetin trimer, with its glycosidic forms; and kaempferol. They were present in large amounts, and this related to the strong biological properties of onion peels [18,19,20].

In my previous publication [20], the fragmentation pattern of the detected metabolites using HPLC/MS to confirm the proposed structures belonged to the classes of hydroxybenzoic acids, anthocyanins, and flavonols. Moreover, isomers of dihydroxybenzoic acid conjugated with hexose and condensation products of quercetin to dihydroxybenzoic acid were also detected. An in-depth analysis of tandem MS data made it possible to preliminarily predict novel phenolic constituents, the presence of which in onion peels could contribute to its bioactivity. The most prominent compounds were protocatechuic acid, cyanidin 3-O-glucoside, (2-(3,4-dihydroxyben zoic)-2,4,6-trihydroxy-3(2H)-benzofuranone, and a condensation product of quercetin to dihydroxybenzoic acid.

In this regard, the significant potent effect of RO-S fraction was mostly assigned to its high quercetin content. These results are in conformity with research previously reported about quercetin, namely, that it exerts a considerable anti-biofilm effect via bacterial-growth or motility inhibition, pathogenicity, and biofilm formation suppression [21]. In conclusion, quercetin is the key flavonoid behind all the health benefits and the significant antibiofilm activity [11,22,23,24]. The presence of quercetin aglycone in higher amounts in the most active tested samples drove us to quantify its content via HPLC analysis.

Finally, the powerful anti-biofilm activity of the red onion waste suggests its use as a natural antibiofilm agent and prompts further investigation into its antimicrobial properties in combination with other antibiotics.

## 4. Materials and Methods

### 4.1. Plant Material and Chemicals

The dry outer red onions scales (*Allium cepa* L.) were collected and purchased from a local market (Giza, Egypt). Quercetin standard solution (95% purity), diethyl ether, trifluoroacetic acid, HPLC-grade methanol, and HPLC-grade acetonitrile were purchased from Sigma–Aldrich Chemical Co. (St. Louis, MO, USA). Hydrochloric acid (HCL) and absolute ethanol (95%) were purchased from the PioChem company (Cairo, Egypt). Vancomycin^®^ vials (500,000 IU) were purchased from Sigma-Tec (SIGMA Pharmaceutical Industries, St. Louis, MO, USA). Carbopol 934 P was provided by Chemical Industries Development Company (Giza, Egypt). Triethanolamine was supplied by Morgan Chemicals Ind. Co. (Cairo, Egypt).

### 4.2. Extraction and Fractionation Procedure

Infection-free red onion outer dry scales were separated from the onion’s bulbs. After grinding, the powdered red onion scales were exhaustively extracted by 1% hydrochloric acidified 70% ethanol by homogenization. Then, the solvent was evaporated under vacuum at a temperature not exceeding 40 °C. The obtained ethanolic extract residue was fractionated to obtain anthocyanin- and flavonoid-rich fraction using the diethyl ether precipitation method. The acidified ethanolic residue (20 g) was dissolved in 200 mL 95% ethanol and then a 10-fold volume of diethyl ether was added in a tightly sealed flask. After 24 h of standing in a refrigerator, the supernatant was decanted, and the red sediment was dried. This sediment was once more dissolved in 100 mL 95% ethanol, and a further 10-fold volume of diethyl ether was added. The supernatant was again decanted and added to the previous volume then evaporated under vacuum, and the red residue was dried in a desiccator for 24 h. The supernatant residue represents the flavonoid-rich fraction, while the red sediment represents the anthocyanin-rich fraction [25]. All samples were stored in a freezer (at −30 °C) until used.

### 4.3. Screening of the Antibacterial Activity

The antibacterial activity of the tested extract/fractions was determined against the following highly virulent bacterial strains: methicillin-resistant *Staphylococcus aureus* (MRSA USA300), *Acinetobacter baumannii AB5075*, *Escherichia coli ATCC8*, and *Pseudomonas aeruginosa PAO1* [26,27,28]. The broth microdilution method was used according to the guidelines of the Clinical and Laboratory Standards Institute to determine the minimum bactericidal concentration (MBC) [29,30]. The experiment was done three independent times and the MBC was recorded as mean ± standard deviation.

### 4.4. Effect of the Tested Extracts on the Biofilm Activity

The anti-biofilm activity of the tested extract/fractions was investigated. They were tested at concentrations below the determined MBC. The biofilm-forming bacterial strains methicillin-resistant *Staphylococcus aureus* (MRSA USA300) and *Acinetobacter baumannii AB5075* were used for this investigation [31].

### 4.5. Biofilm Inhibition Assay

The biofilm inhibition assay was performed using flat-bottom 96-well plates as reported before [31,32]. Different concentrations (sub-MBC) of the tested extract/fractions were tested. Untreated wells (no extract was added) were used as control (100% reference values). The experiment was repeated three independent times. The biofilm inhibition % was calculated using the following equation [31]:$$\mathrm{Biofilm}\;\mathrm{inhibition}\;\%=\frac{\mathrm{OD}\;\mathrm{Control}-\mathrm{OD}\;\mathrm{Test}}{\mathrm{OD}\;\mathrm{Control}}\times 100$$

### 4.6. Biofilm Eradication Assay

The ability of the tested extract/fractions to eradicate the previously established biofilm was investigated. The experiment was performed as reported before [31,32]. Different concentrations (sub-MBC) of the tested extract/fractions were prepared in fresh media and were added to the biofilm plate. Nothing was added to the biofilm control wells (untreated biofilm, 100% reference value). The experiment was repeated three independent times. The biofilm eradication % was calculated using the following equation [31]:$$\mathrm{Biofilm}\;\mathrm{eradication}\;\%=\frac{\mathrm{OD}\;\mathrm{Control}-\mathrm{OD}\;\mathrm{Test}}{\mathrm{OD}\;\mathrm{Control}}\times 100$$

### 4.7. In Vivo MRSA Vaginal Colonization Model

The tested extract/fractions were formulated into Carbopol hydrogel to assess their in vivo antibacterial activity [33]. Each extract was tested at two concentrations: 50 mg/mL and 12.5 mg/mL. All the animal-related procedures were done following the guidelines of the Research Ethics Committee of the Faculty of Pharmacy, Cairo University (Ethical Approval# **MP (3121)**). The rat MRSA vaginal-colonization model was performed as reported before [34,35]. Fifty-four Wistar rats (females/144 ± 20 g) were used in this model. The rats were injected intraperitoneally with 0.5 mg estradiol and 0.4 mg dexamethasone 24 h before colonization. Rats were vaginally inoculated with MRSA USA300 (8 × 10^8^ CFU) suspended in phosphate-buffered saline (PBS) for two successive days. The rats were randomly distributed into nine groups (six rats per group, n = 6) 24 h after the second inoculation, and treatments were administered inter-vaginally (200 µL of the hydrogel using a soft plastic tip). Six groups were treated with any of the tested extract/fractions in Carbopol hydrogel either at the high concentration (50 mg/mL) or the low concentration (12.5 mg/mL). The seventh group was treated with the empty Carbopol hydrogel to serve as the vehicle control. The eighth group was treated with vancomycin in Carbopol hydrogel (50 mg/mL) to serve as the positive control. The ninth group served as the negative control group and did not receive any treatment. An additional group (group 10, n = 6) was used as the healthy control group; this group was not infected and did not receive any treatment. One day post-treatment, the vaginal lumen of each rat was swabbed with a sterile swab. The recovered MRSA bacteria in the swabs were serially diluted in PBS and spotted on mannitol salt agar for the viable count, as described before [34]. The results of the tested groups were analyzed. At the end of the experiment, three rats from each group were used for histopathological examination. The whole excised vaginal tissues were preserved in a 10% formalin-saline solution before preparation of the histological sections using the paraffin method technique [34]. The histological sections of the colonized vaginal tissues were compared to that of healthy uncolonized rats.

### 4.8. HPLC Standardization Method

The total ethanolic extract and its flavonoid-rich fraction were dissolved in HPLC-grade methanol (5 mg/mL), filtered through a sterile 0.45 µm membrane filter, and transferred into sealed vials to be subjected to injection for qualitative and quantitative analysis using the Agilent Technologies 1100 Infinity series HPLC system chromatograph equipped with a diode-array detector (DAD). The analysis was performed on an Eclipse XDBC18 column (150 × 4.6 mm, particle size 5 μm) with a C18 guard column (Phenomenex, Torrance, CA), and the column was operated at room temperature. Separation was carried out with a 0.1% trifluoroacetic acid acidified acetonitrile and 0.1% trifluoroacetic acid acidified water gradient elution program at a flow rate of 1 mL/min. It started with 5% acidified acetonitrile, changed to 35% in 7 min, increased to 50% in 10 min, and at last increased to 100% in 2 min. Chromatograms were recorded at 280, 325, and 520 nm.

Establishment of the standard calibration curve of quercetin was achieved by preparing a standard stock solution by dissolving 10 mg of reference-standard quercetin in 10 mL HPLC-grade methanol. The stock solution was then serially diluted with methanol to obtain concentrations equivalent to 31.25–1000 µg/mL. An aliquot (20 µL) of each concentration was subjected to the HPLC system injection under the same conditions adopted for the extract and fraction in triplicates.

AUC (Area under curve) for each quercetin concentration was recorded, and subsequently, the average was plotted against its corresponding concentration.

Occurrence of quercetin in the total ethanolic extract and its flavonoid-rich fraction was ensured via comparison with the quercetin peak and retention time of the standard quercetin chromatogram. Afterwards, by using the following equation that was obtained from the constructed plot, the quercetin amount was calculated:$$Y\;=21.486x\;+424.41,R^{2}=0.996$$

## 5. Conclusions

In brief, red onion outer scales showed efficient in vitro antibacterial effect against four contagious bacterial strains: methicillin-resistant *Staphylococcus aureus* (MRSA USA300), *Acinetobacter baumannii AB5075*, *Escherichia coli ATCC8,* and *Pseudomonas aeruginosa* PAO1. Furthermore, it showed a promising antibiofilm activity against MRSA. The flavonoid-rich fraction and total extract showed the most potent in vitro and in vivo antibacterial and antibiofilm effects, and that is due to the high content of quercetin. Finally, agro-food wastes could be a source of potential candidates used to reduce the burden of multidrug-resistant strains and defeat biofilm-associated infections.

## Figures and Tables

**Figure 1 molecules-28-00355-f001:**
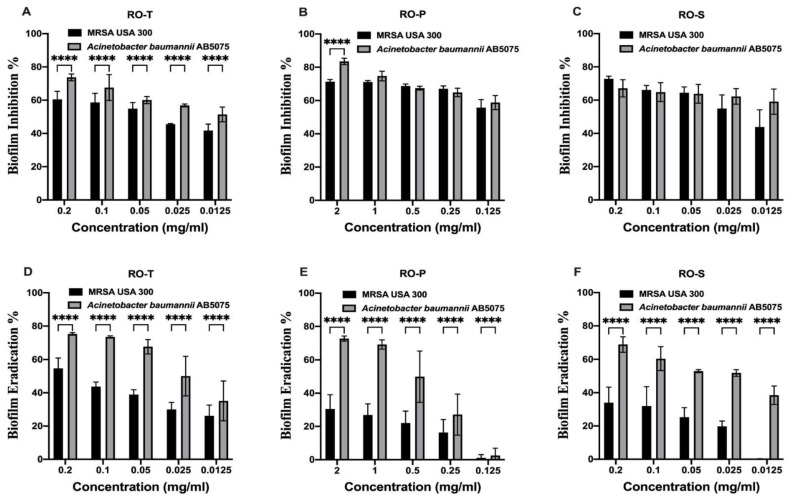
Antibiofilm activity. Effect of various concentrations of extracts RO-T, RO-P, and RO-S on MRSA USA300 and *A. baumannii* AB5057 biofilm formation. Results are expressed as mean biofilm inhibition % ± standard deviation (**A**–**C**) and mean biofilm eradication % ± standard deviation (**D**–**F**). ******** Indicates that the difference is significant at *p* < 0.0001 (Two-way ANOVA, Sidak’s post-hoc test). Methicillin-resistant Staphylococcus aureus (MRSA), the total ethanolic extract of red onion scales (RO-T), anthocyanin-rich fraction (RO-P) and flavonoid-rich fraction (RO-S).

**Figure 2 molecules-28-00355-f002:**
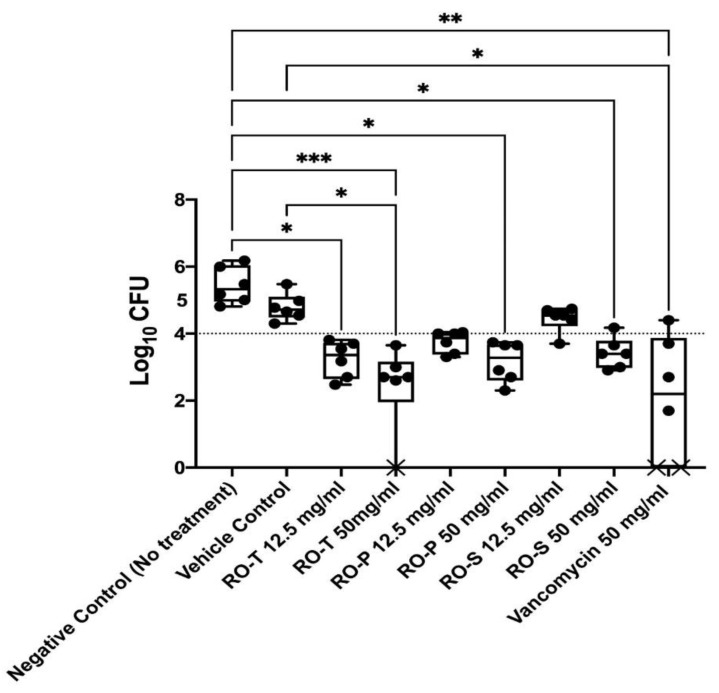
Efficacy of extracts RO-T, RO-P, and RO-S in an in vivo rat model of MRSA vaginal infection. Fifty-four Wistar rats were divided into nine groups (n = 6). Six groups were treated with any of the tested extracts in Carpobol hydrogel, either at the high concertation (50 mg/mL) or the low concertation (12.5 mg/mL). One group was treated with the empty Carbopol hydrogel to serve as the vehicle control. Another group was treated with vancomycin in Carbopol hydrogel (50 mg/mL) to serve as the positive control. Another group served as the negative control group and did not receive any treatment. Each data point in the figure represents a rat. Results are expressed as box plots of bacterial loads recovered from the vaginal lumen. The whiskers span the difference between the minimum and maximum readings, and the horizontal bar represents the median. The symbols *, **, and *** indicate that the difference is significant at *p* < 0.05, < 0.01, and < 0.001, respectively (Kruskal–Wallis Test, Dunn’s post-hoc test). X means no colonies were detected in the sample.

**Figure 3 molecules-28-00355-f003:**
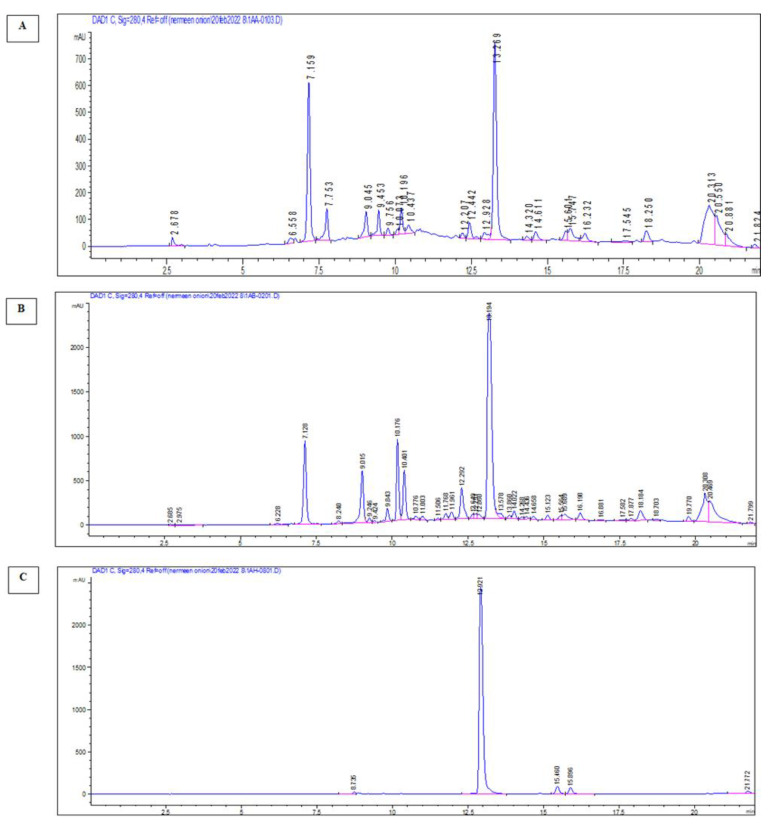
HPLC chromatograms of total extract (**A**), flavonoid-rich fraction (**B**), and quercetin std (**C**) at 280 nm.

**Table 1 molecules-28-00355-t001:** Antibacterial activity of the tested extracts.

	Minimum Bactericidal Concentration (MBC) mg/mL *			
	*MRSA USA300*	*Acinetobacter baumannii AB5075*	*Escherichia coli ATCC87*	*Pseudomonas aeruginosa PAO1*
**RO-T**	0.651 ± 0.226	4.688 ± 2.706	9.375 ± 5.413	16.667 ± 7.217
**RO-P**	4.167 ± 1.804	6.25 ± 0	25 ± 0	#
**RO-S**	0.325 ± 0.113	0.651 ± 0.226	7.292 ± 4.774	12.5 ± 0

* Data are presented as mean ± standard deviation. # No antibacterial activity was detected against the tested organism.

**Table 2 molecules-28-00355-t002:** The half maximal inhibitory/eradication concentration (IC50/EC50) of the tested extracts against biofilm of methicillin-resistant *Staphylococcus aureus* (MRSA USA300) and *Acinetobacter baumannii* AB5075.

		Biofilm Inhibition			Biofilm Eradication		
		Log IC_50_		IC_50_ (mg/mL)	Log EC_50_		EC_50_ (mg/mL)
		mean	SE	mean	mean	SE	mean
**RO-T**	**MRSA USA300**	−1.195	0.164	0.064	−1.797	0.187	0.016
	***Acinetobacter baumannii* AB5075**	−0.818	0.173	0.152	−0.761	0.224	0.174
**RO-P**	**MRSA USA300**	0.311	0.153	2.048	−1.573	0.305	0.027
	***Acinetobacter baumannii* AB5075**	0.476	0.189	2.990	−0.818	0.303	0.152
**RO-S**	**MRSA USA300**	−0.841	0.185	0.144	−2.492	0.309	0.003
	***Acinetobacter baumannii* AB5075**	−0.833	0.139	0.147	−1.092	0.180	0.081

## Data Availability

Not applicable.

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
