# Peer review of "In Vitro and In Vivo Antibiofilm Activity of Red Onion Scales: An Agro-Food Waste"

_molecules, 2023, doi:10.3390/molecules28010355_

Round 1

Reviewer 1 Report

The authors test the antimicrobial activity of extracts from red onion scales for several bacterial pathogens both in vitro and in vivo. The in vitro assays include MIC and inhibition or destruction of biofilms. The in vivo assay uses a rat vaginal colonization model. HPLC was used to characterize the antimicrobial extracts and correlate the antimicrobial activity with quercetin content.

The manuscript displays several strengths related to significance, objectives, and approach. Resistance to currently available antimicrobials is a growing concern. The authors used several bacterial species that are important pathogens and for which antimicrobial resistance has been demonstrated in clinical settings. The authors identify red onion scale extracts and compounds that display antimicrobial activity in relevant in vitro and in vivo assays. The authors include a vehicle negative control and vancomycin positive control for the rat vaginal colonization assay. The data presented generally support the authors interpretations and conclusions. Finally, red onions are an intensively farmed crop and their waste products are abundant.

A notable weakness of the manuscript is the enormous space (fully 10 pages of the 20 page manuscript) devoted to histopathology micrographs of rat vaginal samples that are purely descriptive and non-quantitative, and do not appear to be important for the study. The authors are not establishing a new model of MRSA vaginal colonization. The manuscript is focused on antimicrobial activity but the micrographs are not focused on the bacteria or bacterial biofilms but just on host tissue, sometimes showing evidence for host responses. The micrographs do not add to the quantitation of vaginal colonization or assessment of antimicrobial activity that is performed separately by plating for CFU.

A minor point is that though the use of “murine” to describe a rat model is technically accurate, it would be unfamiliar to many readers more accustomed to seeing this descriptor in the context of mouse studies. I suggest replacing “murine” with “rat” throughout the manuscript for this reason.

Author Response

                                                                                                                                                20 Dec, 2022

Manuscript ID:  molecules-2102312

Dear respected Editor             Journal Molecules

I am pleased to submit a revised version of my manuscript entitled as “ In vitro and In Vivo Antibiofilm Activity of Red Onion Scales: An Agro-Food Waste".

Author(s): Nermeen B Ali, Riham A. El-Shiekh, Rehab M. Ashour, Sabah H. Elgayed, Essam Abdel-Sattar *, Mariam Hassan.

The authors would like to thank the reviewers for the excellent and constructive comments that will increase the significance and quality of our manuscript. We addressed all comments and considered point by point as outlined below and all the required corrections were considered and corrected in the manuscript.

Reviewer #1

The authors test the antimicrobial activity of extracts from red onion scales for several bacterial pathogens both in vitro and in vivo. The in vitro assays include MIC and inhibition or destruction of biofilms. The in vivo assay uses a rat vaginal colonization model. HPLC was used to characterize the antimicrobial extracts and correlate the antimicrobial activity with quercetin content.

The manuscript displays several strengths related to significance, objectives, and approach. Resistance to currently available antimicrobials is a growing concern. The authors used several bacterial species that are important pathogens and for which antimicrobial resistance has been demonstrated in clinical settings. The authors identify red onion scale extracts and compounds that display antimicrobial activity in relevant in vitro and in vivo assays. The authors include a vehicle negative control and vancomycin positive control for the rat vaginal colonization assay. The data presented generally support the authors interpretations and conclusions. Finally, red onions are an intensively farmed crop and their waste products are abundant.

Response: Thank you so much for your valuable comments which are addressed below and considered in the manuscript.

  • A notable weakness of the manuscript is the enormous space (fully 10 pages of the 20-page manuscript) devoted to histopathology micrographs of rat vaginal samples that are purely descriptive and non-quantitative, and do not appear to be important for the study. The authors are not establishing a new model of MRSA vaginal colonization. The manuscript is focused on antimicrobial activity, but the micrographs are not focused on the bacteria or bacterial biofilms but just on host tissue, sometimes showing evidence for host responses. The micrographs do not add to the quantitation of vaginal colonization or assessment of antimicrobial activity that is performed separately by plating for CFU.

Response: Thank you for your valuable comment where the mentioned part was added as supplementary data of the manuscript.

  • A minor point is that though the use of “murine” to describe a rat model is technically accurate, it would be unfamiliar to many readers more accustomed to seeing this descriptor in the context of mouse studies. I suggest replacing “murine” with “rat” throughout the manuscript for this reason.

Response: Corrected.

Thank you very much for considering our manuscript.

Sincerely,

Essam Abdel-Sattar, PhD

Reviewer 2 Report

The authors prepared extracts from onion shells and used it in antimicrobial tests. In my opinion, this is rather routine work with low originality and significance.

It would be more interesting, if the authors did more research on the composition of the extract (not just establishing the content of quercetin) and characterized the compounds associated with the activities and/or showed if there is some synergic effect of having them together.

Also, it would be necessary to show the source of onion and perhaps comparison with onion from elsewhere.

In my opinion, this article is not suitable for publishing in molecules without significant improvements.

Author Response

                                                                                                                                                20 Dec, 2022

Manuscript ID:  molecules-2102312

 Dear respected Editor             Journal Molecules

I am pleased to submit a revised version of my manuscript entitled as “ In vitro and In Vivo Antibiofilm Activity of Red Onion Scales: An Agro-Food Waste".

Author(s): Nermeen B Ali, Riham A. El-Shiekh, Rehab M. Ashour, Sabah H. Elgayed, Essam Abdel-Sattar *, Mariam Hassan.

The authors would like to thank the reviewers for the excellent and constructive comments that will increase the significance and quality of our manuscript. We addressed all comments and considered point by point as outlined below and all the required corrections were considered and corrected in the manuscript.

Reviewers’ comments

 Reviewer #2

The authors prepared extracts from onion shells and used it in antimicrobial tests. In my opinion, this is rather routine work with low originality and significance.

In my opinion, this article is not suitable for publishing in molecules without significant improvements.

  • It would be more interesting, if the authors did more research on the composition of the extract (not just establishing the content of quercetin) and characterized the compounds associated with the activities and/or showed if there is some synergic effect of having them together.

Response: In my previous publication

Abouzed, T. K., del Mar Contreras, M., Sadek, K. M., Shukry, M., Abdelhady, D. H., Gouda, W. M., ... & Abdel-Sattar, E. (2018) ``Red onion scales ameliorated streptozotocin-induced diabetes and diabetic nephropathy in Wistar rats in relation to their metabolite fingerprint. Diabetes research and clinical practice, 140, 253-264``,

An in-depth analysis of tandem MS of red onion scales was published and we predicted several phenolic structures, whose presence in ROS could contribute to its bioactivity. Where we added a paragraph (Lines 126-135) in our discussion part.

  • Also, it would be necessary to show the source of onion and perhaps comparison with onion from elsewhere.

Response: We designed our study to develop an antibiofilm agent from waste products (collected from general vegetable local market), which is matched to the scope of special issue of the journal tracing the biological importance of the Agrofood wastes. Also, a simple method for analysis was developed to standardize red onion scales collected from different sources. The bioactive red Onion scales extract, and fractions were conducted, and results included revealing for the first time the potential of the extract against MRSA vaginal colonization model. The authors have agreed to conduct other detailed comparative studies in the next study on the different available cultivars in Egypt.

Thank you very much for considering our manuscript.

Sincerely,

Essam Abdel-Sattar, PhD

Reviewer 3 Report

This manuscript investigated the in vitro and in vivo antibiofilm activities of acidified ethanolic extract of red onion scales and its fractions against MRSA vaginal colonization model. This is an interesting and meaningful study, and the content is well developed. Therefore, I recommend that this paper could be accepted in present form.

Author Response

                                                                                                                                                20 Dec, 2022

Manuscript ID:  molecules-2102312

Dear respected Editor             Journal Molecules

I am pleased to submit a revised version of my manuscript entitled as “ In vitro and In Vivo Antibiofilm Activity of Red Onion Scales: An Agro-Food Waste".

Author(s): Nermeen B Ali, Riham A. El-Shiekh, Rehab M. Ashour, Sabah H. Elgayed, Essam Abdel-Sattar *, Mariam Hassan.

The authors would like to thank the reviewers for the excellent and constructive comments that will increase the significance and quality of our manuscript. We addressed all comments and considered point by point as outlined below and all the required corrections were considered and corrected in the manuscript.

Reviewers’ comments

Reviewer #3

This manuscript investigated the in vitro and in vivo antibiofilm activities of acidified ethanolic extract of red onion scales and its fractions against MRSA vaginal colonization model. This is an interesting and meaningful study, and the content is well developed. Therefore, I recommend that this paper could be accepted in present form.

We would like to thank the reviewer for its reply.

Thank you very much for considering our manuscript.

Sincerely,

Essam Abdel-Sattar, PhD

Round 2

Reviewer 2 Report

Since my biggest concern was if the article is suitable for this journal (I would expect some more biologically oriented one) and the other reviewers think it is allright, I am also going to support this article to get published here. 

Author Response

Reviewers’ comments

Reviewer #2 (round 2)

Since my biggest concern was if the article is suitable for this journal (I would expect some more biologically oriented one) and the other reviewers think it is all right, I am also going to support this article to get published here. 

Response: The authors would like to thank the reviewer for his comments and for agreeing to consider our paper for publication

Thank you very much